# The Tat Protein of HIV-1 Prevents the Loss of HSV-Specific Memory Adaptive Responses and Favors the Control of Viral Reactivation

**DOI:** 10.3390/vaccines8020274

**Published:** 2020-06-04

**Authors:** Francesco Nicoli, Eleonora Gallerani, Mariaconcetta Sicurella, Salvatore Pacifico, Aurelio Cafaro, Barbara Ensoli, Peggy Marconi, Antonella Caputo, Riccardo Gavioli

**Affiliations:** 1Department of Chemical and Pharmaceutical Sciences, University of Ferrara, 44121 Ferrara, Italy; eleonora.gallerani@unife.it (E.G.); scrmcn@unife.it (M.S.); salvatore.pacifico@student.unife.it (S.P.); mcy@unife.it (P.M.); cpa@unife.it (A.C.); gvr@unife.it (R.G.); 2AIDS National Center, Istituto Superiore di Sanità, 00161 Rome, Italy; aurelio.cafaro@iss.it (A.C.); barbara.ensoli@iss.it (B.E.)

**Keywords:** biologically active HIV-1 Tat protein, HSV-1 infection, HSV-1 immune responses, persistence of HSV-immune memory

## Abstract

The development of therapeutic strategies to control the reactivation of the Herpes Simplex Virus (HSV) is an unaddressed priority. In this study, we evaluated whether Tat, a HIV-1 protein displaying adjuvant functions, could improve previously established HSV-specific memory responses and prevent viral reactivation. To this aim, mice were infected with non-lethal doses of HSV-1 and, 44 days later, injected or not with Tat. Mice were then monitored to check their health status and measure memory HSV-specific cellular and humoral responses. The appearance of symptoms associated with HSV-reactivation was observed at significantly higher frequencies in the control group than in the Tat-treated mice. In addition, the control animals experienced a time-dependent decrease in HSV-specific Immunoglobulin G (IgG), while the Tat-treated mice maintained antibody titers over time. IgG levels were directly correlated with the number of HSV-specific CD8^+^ T cells, suggesting an effect of Tat on both arms of the adaptive immunity. Consistent with the maintenance of HSV-specific immune memory, Tat-treated mice showed a better control of HSV-1 re-infection. Although further studies are necessary to assess whether similar effects are observed in other models, these results indicate that Tat exerts a therapeutic effect against latent HSV-1 infection and re-infection by favoring the maintenance of adaptive immunity.

## 1. Introduction

The development of preventive and therapeutic vaccines against Herpes Simplex Virus (HSV) is a global health priority for many reasons: (1) genital herpes affects half a billion people between the ages of 15 and 49 worldwide [1,2], causing diseases which are particularly severe and potentially lethal in newborns and immunocompromised hosts; (2) HSV infection is one of the worldwide leading causes of encephalitis, with high hospitalization and mortality rates when compared with other etiologic causes, especially in immunosuppressed and elderly hosts [3,4]; (3) HSV-1 and 2 infections have also been associated with the onset and morbidity of different non-communicable diseases, such as Alzheimer’s disease [5] and atherosclerosis [6] and with an increased risk of horizontal and vertical HIV acquisition and transmission [7,8] and bacterial vaginosis [9]. 

Current HSV drugs are only efficacious against replicating HSV; they do not abrogate neuronal latency nor do they eradicate the virus. Drug resistance has also been reported at high levels in immunocompromised patients leading to severe outcomes, including encephalitis and death [10,11]. Thus, the only means to prevent HSV infection, virus spreading and reactivation from latency is the use of vaccines against HSV, as also prioritized by WHO [12]. However, despite over 60 years of intensive research, HSV vaccines—preventive or therapeutic —are not yet available [13]. 

Adaptive immunity is considered essential for long-lasting HSV control. CD8^+^ T cells found at both mucosal sites [14] and trigeminal ganglia [15] and in particular those with an effector memory (EM) phenotype, are directly responsible for the control of ocular and vaginal herpes [16,17,18,19] and of reactivation from latency [20]. Humoral responses have been shown to contribute to protection, although they are insufficient alone [16,21,22,23]. Therefore, strategies aimed at boosting HSV-specific pre-existing immunity (memory lymphocytes) may be beneficial to prevent viral reactivation. Memory lymphocytes are long-lived cells that, if necessary, will give rise to secondary responses, quicker and more potent than primary ones. However, immune memory physiologically wanes due to several causes, including aging [24,25,26] and pathological conditions such as transplantations [27], immunodeficiency [28] and immunosuppressive infections [29]. In addition, the protection conferred by memory recall responses is influenced by the heterogeneity of the memory pool [30,31]. However, this heterogeneity is lost with physiological aging or after certain infections [32,33,34,35], resulting in the accelerated loss of protection [24,25,26,36]. Therefore, the development of strategies aimed at improving the maintenance and recall capacity of memory adaptive responses may be key for the long-term control of HSV.

In previous studies aimed at developing a novel preventive vaccine against HSV, we generated a HSV-based vector that expressed the HIV-1 Tat protein, a potent immune modulator [23,37,38] that can increase and broaden protective humoral and cellular immunity against HSV thanks to its intrinsic immunomodulatory properties that broaden and maintain CD8^+^ T cell responses against intracellular pathogens while increasing anti-HSV1 antibody responses in mice [16,23]. We therefore wondered if Tat could also improve previously established HSV-specific memory responses, thus preventing viral reactivation. 

## 2. Materials and Methods

### 2.1. Peptides and Antibodies

The biologically active HIV-1 Tat protein (HTLV-IIIB isolate, BH10 clone), provided by Diatheva, was produced in *Escherichia coli,* as previously described [39], and formulated in saline buffer in the presence of 1% saccarose and 1% human serum albumin and stored at −80 °C. The HSV-1 K^d^-restricted SSIEFARL (SSI) peptide, derived from glycoprotein B and corresponding to an immunodominant CTL epitope, was used to evaluate T cell responses in C57BL/6 mice, as previously described [16]. Anti-Tat polyclonal (ANT0001) and monoclonal (NT3 2D1.1) antibodies were purchased, respectively, from Diatheva (Diatheva, Fano, Italy) and the NIH Research and Reference Reagent Program (German Town, MD, USA). 

### 2.2. Herpes Simplex Virus Type 1 and Mice

Wild-type HSV type 1 (HSV-1, LV strain) was purified and titrated by the plaque assay method, as previously described [23]. Seven to eight days before intravaginal (IV) inoculation or challenge, female C57BL/6 mice (Charles-River, Lecco, Italy) were injected subcutaneously in the neck with 2 mg/100 µL of Depo-Provera® (Depo-medroxy-progesterone acetate; Pharmacia & Upjohn). IV infection, with 10^3^ or 10^4^ plaque forming units (PFU) of HSV-1, and IV challenge, with 10^7^ PFU of HSV1, were performed as previously described [23]. The experiment with 10^3^ PFU was performed with 10 animals. The experiment with 10^4^ PFU was performed twice with 32 and 12 animals, respectively. After HSV-1 infection and challenge, mice were observed daily to monitor the appearance of local and/or systemic clinical signs of infection including death. Disease signs were classified as ruffled hair (score = 1), cold sores (score = 2), limb paralysis (score = 3) and death (score = 4). Blood samples for detection of HSV1-specific immune responses were collected from the retro-orbital plexus. At day 44 post-infection (p.i.), mice were mixed and randomly assigned to receive Tat or buffer. Before day 44, the infection was asymptomatic or mildly symptomatic (score = 1) in the majority of mice, and less than the 10% of the animals developed vaginal lesions (cold sores). All animal experiments were conducted in conformity to European and Institutional guidelines as ruled by the Italian Ministry of Health.

### 2.3. Determination of Cellular and Humoral Responses

Characterization of the number and phenotype of HSV-specific CD8^+^ T cells specific to the SSI peptide was performed by flow cytometry using dextramers (Immudex, Copenhagen, Denmark), as previously described [16]. The following antibodies were used: PerCP-Cy5.5 anti-CD3 (TONBO Biosciences, Societa Italiana Chimici Rome, Italy); APC anti-CD62L (Immunotools, Friesoythe, Germany); BV510 anti-CD44 (Biolegend, Campoverde S.r.l. Milano, Italy) and APC-H7 anti-CD8 (Becton Dickinson Milano, Italy). Samples were acquired on FACS Aria flow cytometer (BD) within 2 h of fixation. Flow cytometry data were analyzed using FlowJo (version 9.5.3; Tree Star Inc., Ashland, OR, USA). 

Sera for antibody determinations were collected, stored and assessed by using the ELISA test for the presence and titers of anti-HSV IgG, as previously described [23].

### 2.4. Statistics

Statistical analyses were performed using Prism software (GraphPad, San Diego, CA, USA). Significance was assigned at *p* < 0.05. The Kaplan–Meier test was used to estimate the probability of clinical manifestations. The magnitude of disease scores after challenge and of cellular responses were analyzed using the two-tailed Mann–Whitney test after having assessed that data were not normally distributed (Kolmogorov–Smirnov test). The kinetics of humoral responses were compared over time in the same animals through a paired Student’s t test after having assessed that data were normally distributed (Kolmogorov–Smirnov test).

## 3. Results and Discussion

### 3.1. The HIV-1 Tat Protein Has a Therapeutic Effect in Mice Infected with HSV-1

Our previous studies have indicated that the simultaneous administration of the Tat protein with heterologous antigens improves both cellular and humoral immune responses against the antigens in in vitro and murine models [23,38]. However, in in vitro experiments, this effect was not observed when Tat was added to T cells after priming (i.e., during the expansion phase of the immune response) [40]. In agreement with these results, the administration of Tat to mice previously infected with HSV-1 7 days before, did not improve immune responses nor protection against a lethal challenge (data not shown). Since it has also been demonstrated in vitro that Tat exerts different effects on T cells depending on their activation status [41], we wondered whether Tat may affect the pre-existing adaptive memory immune responses. To test this hypothesis, C57BL/6 mice were infected intravaginally (IV) with a low dose (10^4^ PFU) of wild-type HSV-1. At day 44 p.i., 5 µg of the HIV-1 Tat protein—or, in the control group, the equivalent volume of the buffer alone—was administered intradermally (ID) to mice (Figure 1A). Animals were then monitored up to day 108 p.i. (day 64 after Tat administration). Interestingly, we noticed a spontaneous appearance of signs such as vesicles and/or hair loss around the genital area, starting a few days after treatment. These disease signs, compatible with a viral reactivation, were significantly more frequent in controls than in Tat-treated mice (Figure 1B). Similar results were found when Tat was administered to mice previously infected with a lower dose of HSV-1 (10^3^ PFU) (Appendix A). Further, all Tat-treated mice developed significantly milder signs of infection after an IV challenge with a high dose of HSV-1 (10^7^ PFU) administered on day 108 post-infection. The difference in disease symptoms was particularly evident between 7 and 9 days after challenge before the disease signs spontaneously healed in both groups (Figure 1C).

Previous reports have shown that the transduction domain of Tat exhibits antiviral activities [42,43]. However, in our experiments we rule out a direct effect of Tat on HSV-1 virions, which might only occur in the very remote case of an ongoing subclinical reactivation at the time of treatment (day 44). Instead, we asked whether anti-Tat specific antibodies (Ab) could recognize and cross-react with HSV-1. Although anti-Tat Ab are detectable in only a fraction of HIV-infected individuals [44,45], they are effectively induced after administration of the Tat protein in humans [46,47,48] and animals [49,50,51]. Consistent with this, 100% of Tat-treated mice displayed anti-Tat IgG at both 60 and 108 days p.i. (16 and 64 days after Tat inoculation, respectively), with mean titers in the range of 10^4^ (data not shown). To evaluate whether anti-Tat Ab hamper HSV-1 infectivity, an in vitro plaque reduction assay was performed by pre-incubating HSV-1 with sera of Tat- and/or HSV-1-immune mice. Anti-Tat Ab did not prevent or reduce HSV-1 infection of Vero cells, in contrast to what was observed using anti-HSV-1 immune sera, regardless of the presence of anti-Tat IgG (Appendix A), thus excluding non-specific effects of anti-Tat Ab on HSV-1 infectivity. Overall, these results indicate that the Tat protein exerts a therapeutic effect against latent HSV-1 infection and re-infection.

### 3.2. Administration of the HIV-1 Tat Protein Prevents the Time-Dependent Reduction in Antigen-Specific Adaptive Humoral Responses

We next assessed whether Tat may affect already established anti-HSV-1 immune responses. To this aim, the amount of circulating CD8^+^ T cells specific to the HSV-1 K^d^-restricted SSI epitope were evaluated, since they play an important role in protection from HSV-1 infection [16,23]. Although not statistically significant, increased numbers of SSI-specific CD8^+^ T cells were observed in the blood of Tat-treated mice compared to that of control animals both at early (day 60 p.i.) and late (day 108 p.i.) time points after administration of Tat (Figure 2A). Previously, we have reported that Tat favors the late differentiation of T cells [38,40,52]. In addition, it has been shown that differentiated (e.g., effector memory) T cells are important correlates of protection in HSV-1 infection [16,17,18,19,53]. Thus, we characterized the phenotype of blood circulating SSI-specific CD8^+^ T cells. Consistent with the mild signs of disease observed in mice previously infected with 10^4^ PFU and then challenged with 10^7^ PFU of HSV-1 (Figure 1C), all animals showed a high proportion (60–80%) of effector memory cells (EM) within SSI-specific CD8^+^ T cells, regardless of their group (Figure 2B). In the very same mice, the proportion of EM cells within the whole CD8 compartment (regardless of the antigen specificity) was approximately 10–15% and directly correlated with the percentage of SSI-specific CD8^+^ T cells (*p* < 0.0001) (Appendix A), suggesting that HSV-specific CD8^+^ T cells affect and drive the size of the EM subset. The frequency of EM SSI-specific CD8^+^ T cells, although slightly decreased over time, persisted at levels of above 60% in both groups (Figure 2B). Therefore, the long-term cellular response against HSV is mainly of EM type and Tat did not change this pattern. Considering the importance of EM T cells for local responses, it will be important, in future experiments, to measure the amount and phenotype of HSV-specific CD8^+^ T cells in mucosal tissues.

Surprisingly, while control mice experienced a physiological, time-dependent, decrease in HSV-specific IgG, the same was not found in Tat-treated mice which, instead, showed similar levels of HSV-specific IgG up to 108 days p.i. (Figure 2C). This indicates that Tat prevents the waning of circulating antibodies against a heterologous antigen such as HSV-1. It will require further research to determine whether this phenomenon is due to a direct effect of Tat on B cells or is T cell-dependent.

Very few studies, which are usually focused in the context of HIV-associated lymphomas [54,55,56], have investigated the direct effects of Tat on B cells. Instead, it has been widely reported that Tat exhibits potent immunomodulatory functions on antigen presenting cells [57,58], which may therefore be more prone to presenting antigens and providing costimulation signals in HSV reactivation. In addition, Tat can influence T-cell programming [38,40,52] and prevent apoptosis in T helper lymphocytes [59,60,61,62], whose role in maintaining and recalling humoral responses has also been demonstrated [63,64,65,66]. In line with the hypothesis of a strong interaction between cellular (CD8 and CD4) and humoral responses, we observed a direct correlation between HSV-specific IgG levels and the percentage of circulating SSI-specific CD8^+^ T cells (*p* = 0.009) (Appendix A). Therefore, we may speculate that Tat effects on the T-cell compartment also influence humoral immunity. This phenomenon is shared by several pathogen-associated molecular patterns that, by modulating antigen presentation and costimulation to helper T cells, affect both cytotoxic and humoral responses [67,68]. Indeed Tat, further to inducing the release of IL-6 [69,70], which is important for B-cell maturation and T cell-dependent humoral responses [71,72], binds the toll-like receptor (TLR4) [73]. TLR4 signaling profoundly influences the functions of T and B lymphocytes [74,75,76] and may also rewire the metabolic programs of lymphocytes towards lipid usage, thus potentially impacting the persistence and functionality of memory cells, which are highly reliant on fatty acid oxidation [31,75]. We recently reported that Ab induced by a vaccine adjuvanted with a TLR4 ligand (TLR4L) lasted longer than those elicited by a similar vaccine that did not contain a TLR4L [77]. Although in this work, we administered Tat to mice after the antigen exposure and not simultaneously, we cannot rule out a general effect of TLR4L on plasma cell survival.

The results obtained from human vaccine studies have already shown that HIV-infected subjects vaccinated with Tat can improve memory adaptive responses to heterologous antigens [46]. The data presented here confirm this observation and also suggest that this phenomenon is probably due not only to the neutralization of the deleterious effects of Tat on HIV-infected subjects but also to the direct support on the maintenance of adaptive immunity mediated by Tat. Together, these data indicate that Tat can influence already established memory responses, preventing their physiological decline.

## 4. Conclusions

In this report, we have shown that the Tat protein of HIV exerts therapeutic effects in animals with a previously acquired HSV-1 infection. Our observations suggest that these effects are mediated by the maintenance of adaptive immunity, and in particular humoral responses, although the molecular mechanism remains unknown. Recently, non-specific effects exerted by some vaccines against heterologous antigens have been described, and this phenomenon was named “trained immunity” [78,79,80]. It is important to further investigate whether Tat displays similar mechanisms and whether its effects may also be exerted on other antigens. In addition, further studies on innate immune cells may help to identify the mechanism. However, these findings provide important clues to the use of Tat for the control of latent HSV-1.

## Figures and Tables

**Figure 1 vaccines-08-00274-f001:**
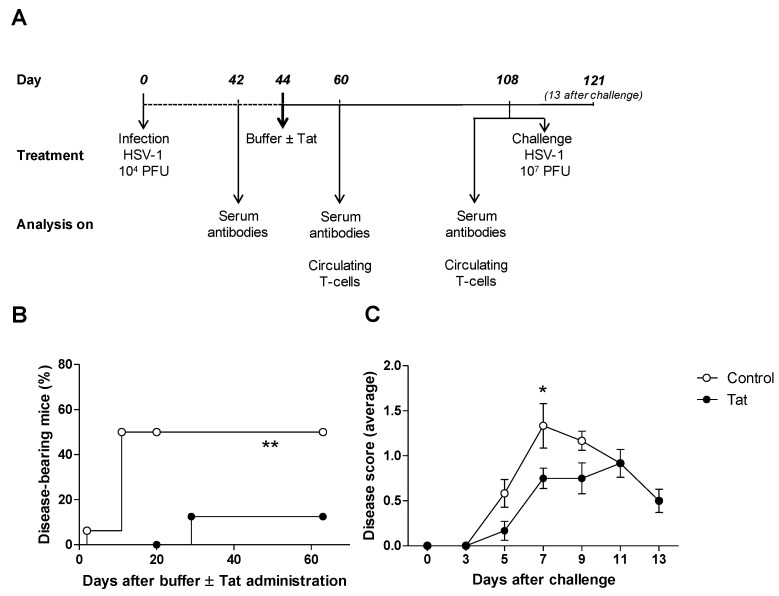
Tat exerts therapeutic effects on previously acquired Herpes Simplex Virus (HSV-1) infection. (**A**) Schematic representation of the experimental protocol and analysis. C57BL/6 mice were infected by the intravaginal (IV) route with 10^4^ plaque forming units (PFU)/mouse of HSV-1 and, on day 44 post-infection (p.i.), treated with HIV-1 Tat (5 µg) administered by the intradermal (ID) route into the back. Controls were given only the Tat buffer. This treatment schedule was repeated in two independent experiments with 16 and 6 mice per group, respectively. Disease scores were monitored up to day 108 p.i. in both experiments. The following treatments and analyses were performed in the second experiment: On day 108 p.i. (i.e., day 64 after buffer ± Tat administration), mice were challenged IV with 10^7^ PFU/mouse of HSV-1 and observed for two weeks after challenge (day 121 p.i.). Mice were bled on days 42 (before Tat treatment), 60 and 108 (before virus challenge) p.i. to analyze humoral and T cell responses. (**B**) Analysis of disease signs after Tat treatment. Mice were observed twice a week after Tat treatment to monitor the appearance of signs of disease. The Kaplan–Meier test was used to estimate the percentage of mice developing clinical manifestation after treatment (*n* = 16 per group). One representative experiment out of two is shown. (**C**) Analysis of signs of disease after HSV-1 challenge. Mice were observed every two days up to day 13 after challenge. For each group, mean disease scores (± SEM) are shown and analyzed statistically using the two-tailed Mann–Whitney test (*n* = 6 per group). * *p* < 0.05, ** *p* < 0.01.

**Figure 2 vaccines-08-00274-f002:**
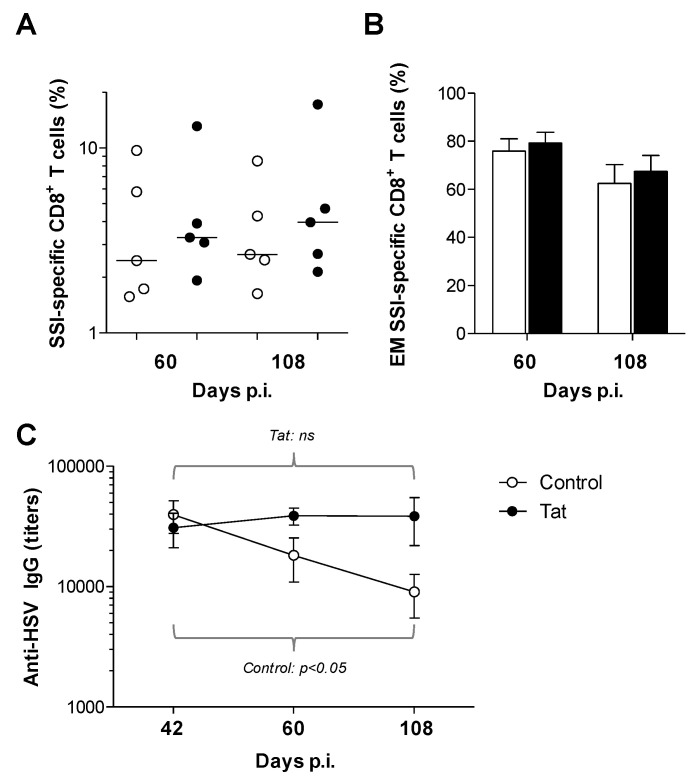
Tat improves the maintenance of previously established immune memory responses. (**A**) Analysis of SSIEFARL (SSI)-specific CD8^+^ T cells. Blood was collected from the retro-orbital bleeding of mice *(n* = 5 per group) at days 60 and 108 p.i. (days 16 and 64 after buffer ± Tat inoculation, respectively) to determine the percentage of SSI-specific CD8^+^ T cells. Dots represent the results of single mice and lines represent median values. (**B**) Analysis of effector memory (EM) SSI-specific CD8^+^ T cells. The percentage of peripheral blood circulating EM (CD44^+^ CD62L^−^) cells within the SSI-specific CD8^+^ T lymphocyte pool was measured at days 60 and 108 p.i. (days 16 and 64 after buffer ± Tat inoculation, respectively). Bars represent the mean ± SEM (*n* = 5 per group). (**C**) Analysis of anti-HSV-1 antibody responses. Sera were collected at days 42, 60 and 108 p.i. (two days before buffer ± Tat treatment, and days 16 and 64 after buffer ± Tat inoculation, respectively) to assess the presence and titers of HSV-specific IgG. Dots represent the mean ± SEM (*n* = 5 per group). A paired Student’s t test was used to estimate the changes over time in HSV-specific IgG titers. n.s. = not significant.

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
