# Peer review of "The Tat Protein of HIV-1 Prevents the Loss of HSV-Specific Memory Adaptive Responses and Favors the Control of Viral Reactivation"

_vaccines, 2020, doi:10.3390/vaccines8020274_

Round 1
Reviewer 1 Report
This is a simple study aimed to evaluate the adjuvant effect of the Tat protein on recalling adaptative response to latent HSV-1 infection.
The authors found that, differently from the vast majority of adjuvants, which enhance primary responses by modulating innate immunity mainly, Tat protein ameliorates HSV-1 infection severity through the maintenance of virus-specific IgG humoral response. However, Tat does not affect significantly the development and/or survival of effector memory CD8 T cells. This does not rule out potential effects on cytolytic activity (here not analyzed)
It is a very interesting finding that paves the way to elucidate the immune mechanisms behind the adjuvant properties for Tat on memory adaptive responses to heterologous antigens.
It seems that Tat acts on ready elements of immune response triggered by the virus or/and the effector program in standby during viral latency.
As authors point, Tat protein might modulate viral antigen presentation enhancing costimulation (i.e by sustained expression of GITRL, B7, etc.) for helper T response and hence, the observed recalling on humoral response.
Moreover, the consequences of the persistence of specific IgG antibodies on virus clearance and reduced disease signs may involve ADCC by NK cells.
The study is mostly well-driven, but the presentation of the results appears to be slightly confusing:
- The time-course of the disease is not too clear. Could the authors include a figure showing the disease incidence and scoring from 0 to 44 days? The selected moment for Tat administration may be clarified.
- The study starts with 16 mice per group (Fig. 1B), but the challenge only was done over 6 animals per group. Similarly, the immune response (EM CD8 T cells and anti-HSV IgG titers) was only analyzed in 5 mice per group.
Is there any reason?. What happened with the rest of the animals?
- The effector memory CD8 T cells were only evaluated in blood. Why not in mucosal sites?
- It seems that Tat protein affects both specific naïve and memory CD8 T cells stimulation, as shown in Figure 2A, but CD8 T response did not evaluate after the HSV-1 challenge. Why?
- In Figure S1, mice were examined twice a week, for 35 days (about 5 weeks). However, the graph only shows 3 time-points… The x-axis scale could be equally fixed for Figure 1B and Figure S1.
- Line 101: “ELISA” instead of Elisa
Author Response
First of all, we would like to thank the Reviewer for the appreciation of our manuscript and
for providing further suggestions on the mechanism that we have now included in the
revised Manuscript (page 6, lines 221-224 and 247-248).
Please find below a point-by-point response to the Reviewer’s comments:
Reviewer (R): The study is mostly well-driven, but the presentation of the results appears to
be slightly confusing:
- The time-course of the disease is not too clear. Could the authors include a figure showing
the disease incidence and scoring from 0 to 44 days? The selected moment for Tat
administration may be clarified.
Authors (A): First of all, we would like to thank the Reviewer for the appreciation of our
manuscript and for providing further suggestions on the mechanism that we have now
included in the revised Manuscript (page 6, lines 221-224 and 247-248).
She/He is right that we did not properly addressed the time-course, neither the status of
mice before Tat administration. This has been now specified in the revised version of
Materials and Methods (pages 2-3, lines 93-95). Regarding the choice of day 44 for the
treatment administration: based on published and unpublished previous experiments, we
identified that, in this disease model, the contraction phase of cellular response ended
about 3 weeks after the primary infection. Therefore, we wanted to wait 3 additional weeks
to inject mice, waiting for having memory cells already established.
R: The study starts with 16 mice per group (Fig. 1B), but the challenge only was done over 6
animals per group. Similarly, the immune response (EM CD8 T cells and anti-HSV IgG titers)
was only analyzed in 5 mice per group.
Is there any reason?. What happened with the rest of the animals?
A: The experiment with 104 PFU was repeated twice, the first time with 16 animals per
group and the second time with 6 animals per group. Disease signs after Tat/buffer
administration were checked in both experiments, with comparable results. Only one
representative experiment out of two is shown in Figure 1, as now stated in the revised
version of the Figure Legend (page 4, line 167).
The immunological measures and the challenge were performed only in the second
experiment (6 mice/group). All mice were challenged, while we could collect enough
amount of blood to perform ELISA and FACS analyses only on 5 animals/group.
We apologize for the confusion and we thank the reviewer for rising this point, which we
tried to clarify in the revised version of Materials and Methods (page 2, lines 87-89) and in
the Figure 1 Legend (page 4, lines 158-161).
Note for the Reviewer: ELISA were performed on few animals per group (n=3) also in the
first experiment. Results were comparable to those later performed on the second
experiment and presented in Figure 2.
R: The effector memory CD8 T cells were only evaluated in blood. Why not in mucosal sites?
A: This is a very relevant observation. Even in our previous experience, we noticed that
effects on T-cells (during a HSV infection) could be tissue-specific. However, we purposely
focused only on blood to keep the mice alive and then challenge them. The importance of
mucosal responses is now discussed in the revised version of the manuscript (page 5, lines
190-192).
R: It seems that Tat protein affects both specific naïve and memory CD8 T cells stimulation,
as shown in Figure 2A, but CD8 T response did not evaluate after the HSV-1 challenge. Why?
A: This work is somehow preliminary and more aspects need to be studied. As Vaccines
gives the possibility to publish “preliminary, but significant, results” in the form of “short
Communication”, we thought that the novelty of our data could fit in this special issue
dedicated to HSV, although some aspects of our research still require further studies.
Indeed, in the set of experiments we performed, the primary endpoint was the assessment
of clinical outcomes; as our previous experience has revealed that the recruitment of T cells
in the vaginal tissue after challenge was crucial for the protection, checking this aspect
would have implied to sacrifice mice, thus not allowing the observation of clinical disease
score after challenge.
R: In Figure S1, mice were examined twice a week, for 35 days (about 5 weeks). However,
the graph only shows 3 time-points… The x-axis scale could be equally fixed for Figure 1B and
Figure S1.
To avoid an overcrowded representation of the results, the Kaplan-Meier curve highlights
only those days where an event was reported, and for this reason, the graphs gave the
impression that only 3 observations were done. In the revised manuscript, we have changed
the style of the Kaplan-Meier curves in both Figures 1 and S1 to a more “classical” one,
hoping that the new figures allow a better comprehension of the results. In the same
figures, the X axis has been changed as suggested. For consistency, we harmonized also the
Y axis. In addition, because of a mistake in compiling the table behind the graph, it appeared
that mice were followed until day 35, while animals were followed up to day 39. We
apologize for this inconvenience.
R: Line 101: “ELISA” instead of Elisa
A: Thanks for pointing this out, the word has been changed (page 3, line 106
Reviewer 2 Report
Nicoli et al investigate the role of the HIV Tat protein in enhancing immune responses in HSV-1 infected mice and alleviating disease. They find that administration of recombinant Tat protein provides a degree of protection. Anti HSV-1 antibody levels were maintained, and an elevation of CD8 T cells was detected though not statistically significant. While intriguing, several issue need to be resolved prior to publication.
The most pressing issue with the manuscript is the lack of methods describing statistical analysis. Why some tests are performed in some instances and not in others is lacking, as well as the software package used for analysis. Any claims made about statistics therefore can’t be analyzed. Why is the Wilcoxon signed rank test used in Supplemental Figure 2? Why a paired Student’s t test to analyze a data set over time in Figure 2C? Figure 2A was not significant, perhaps state the test used and report the p-value, as it looks close? Was an outlier test done (doubtful any would be removed, but worth a try).
Why are the control and Tat data sets combined for Supplemental figure 3 analysis?
There is also an inadequate description of how disease was scored. A list of symptoms is provided but no description of how that gets converted to a number.
I am also confused by lines 171-174. Was there a 10-15% increase in Tat treated animals (compared to controls?) or of antigen-specific cells, which does not appear to be the case? This section needs to be re-written and the data shown as part of figure 2 so the reader can understand the point the authors are trying to make.
Minor issues:
line 45: what are the severe outcomes?
line 45: “thus the only mean” should read “thus the only means”
line 64: “In the attempt of developing novel preventive” needs to be re-worded to make sense
line 65-67: references should be included to support the idea that Tat is an important potent immune modulator
line 85: is “IVag”a typo? if not what does it mean?
line 101: change Elisa to ELISA
line 116: change “buffer used to resuspend Tat” to state “buffer alone”
line 118: was should be changed to were
line 119: “stating few days” does not make sense and should be re-written for clarity
line 194: “they were” is not an appropriate phrase. It could be deleted and the sentence re-worded.
Author Response
We are grateful to the Reviewer for the careful assessment of our manuscript and for raising
relevant issues that helped us to improve the quality of our work.
Please find below a point-by-point response to the Reviewer’s comments:
Reviewer (R): The most pressing issue with the manuscript is the lack of methods describing
statistical analysis. Why some tests are performed in some instances and not in others is
lacking, as well as the software package used for analysis. Any claims made about statistics
therefore can’t be analyzed. Why is the Wilcoxon signed rank test used in Supplemental
Figure 2? Why a paired Student’s t test to analyze a data set over time in Figure 2C? Figure
2A was not significant, perhaps state the test used and report the p-value, as it looks
close? Was an outlier test done (doubtful any would be removed, but worth a try).
Authors (A): The Reviewer is right that a description of, and a rational for, statistics are
missing, and have now been inserted in the revised version of Materials and Methods (page
3, lines 108-115). We are grateful for this comment which helped us to improve the quality
and clarity of our manuscript. In respect to the single issues raised by the Reviewer:
-paired tests were used in Figure S2, as each condition was compared with its own control,
and Figure 2C, as the same mice were analyzed over time.
- an outlier test was performed for Figure 2A but, as rightly pointed out by the Reviewer, we
could not identify any. The p value of the presented results is not close to significance.
R: Why are the control and Tat data sets combined for Supplemental figure 3 analysis?
A: We thank the Reviewer for giving us the possibility to explain this point. We believe that,
most likely, the correlations we observed describe two general phenomena that occur
irrespectively to the type of treatment (CD8 EM subset dominated by HSV-specific T cells,
and linkage between T and B cell responses). However, to provide all necessary elements to
interpret the data, we inserted, in the revised Supplemental Figure 3 legend, the r and p
values for each group.
R: There is also an inadequate description of how disease was scored. A list of symptoms is
provided but no description of how that gets converted to a number.
A: We have clarified this point in the in the revised version of Materials and Methods (page
2, lines 90-92).
R: I am also confused by lines 171-174. Was there a 10-15% increase in Tat treated animals
(compared to controls?) or of antigen-specific cells, which does not appear to be the
case? This section needs to be re-written and the data shown as part of figure 2 so the
reader can understand the point the authors are trying to make.
A: We have revised and clarified this section as suggested by the Reviewer (page 5, lines
184-190 of the revised manuscript). Although we did not merge Figures 2 and S3, we think
that the revision we made will help the reader to understand the points we made.
Minor issues:
R: line 45: what are the severe outcomes?
A: The most severe are encephalitis and death, which have now been specified (line 45,
page 1 of the revised manuscript).
R: line 45: “thus the only mean” should read “thus the only means”
A: Thanks, we have revised this spelling mistake (line 46, page 2 of the revised manuscript).
R: line 64: “In the attempt of developing novel preventive” needs to be re-worded to make
sense
line 65-67: references should be included to support the idea that Tat is an important potent
immune modulator
A: The sentence has been rewritten and references added, as suggested by the Reviewer
(lines 64-65, page 2 of the revised manuscript).
R: line 85: is “IVag”a typo? if not what does it mean?
A: We have revised this spelling mistake (line 85, page 2 of the revised manuscript).
R: line 101: change Elisa to ELISA
A: Thanks for noticing, the word has been corrected (line 106, page 3 of the revised
manuscript).
R: line 116: change “buffer used to resuspend Tat” to state “buffer alone”
A: We have revised the sentence as suggested (line 128, page 3 of the revised manuscript).
R: line 118: was should be changed to were
A: The sentence has been rewritten (lines 130-131, page 3 of the revised manuscript).
R: line 119: “stating few days” does not make sense and should be re-written for clarity
A: “stating” was changed with “starting”, we are sorry for the mistake (line 131, page 3 of
the revised manuscript).
R: line 194: “they were” is not an appropriate phrase. It could be deleted and the sentence
re-worded.
A: We are grateful for the suggestion; the sentence has been rewritten (lines 212-213, page
6 of the revised manuscript).
Round 2
Reviewer 1 Report
The communication has been properly improved.
Reviewer 2 Report
I have no further concerns